# Unveiling Limestone Orchid Hotspots in the Karst Hills of Northern Peninsular Malaysia

**Shahrul Nizam Abu Bakar** [1][iD]**, Farhan Rashid** [1][iD]**, Muhammad Hilmi Jamaluddin** [1]**, Muhamad Faizal Md Azmi** [1]**, Ahmad Sofiman Othman** [1]**, Rahmad Zakaria** [1]**, Azimah Abd Rahman** [2]**, Akmal Raffi** [3] **and Farah Alia Nordin** [1,*][iD]

[1] School of Biological Sciences, Universiti Sains Malaysia, Gelugor 11800, Pulau Pinang, Malaysia; shahrulbakar96@gmail.com (S.N.A.B.); farhanrashidsimon@gmail.com (F.R.); muhdhilmi@usm.my (M.H.J.); faizal1972@gmail.com (M.F.M.A.); sofiman@usm.my (A.S.O.); rahmadz@usm.my (R.Z.)

[2] School of Humanities, Universiti Sains Malaysia, Gelugor 11800, Pulau Pinang, Malaysia; azimahrahman@usm.my

[3] Faculty of Resource Science and Technology, Universiti Malaysia Sarawak, Kota Samarahan 94300, Sarawak, Malaysia; mrmakmal@unimas.my

* Correspondence: farahalianordin@usm.my

**Abstract:** To date, documentation of the diversity of orchids from the limestone hills in the northern part of Peninsular Malaysia, particularly in the states of Kedah and Perak, is still lacking, with limited literature available. There is indeed an urgent need to fill this knowledge gap, so that proper documentation of the diversity of orchids from this unique karst habitat can be prepared. In this study, a series of 12 months of diversity assessments on five limestone hills in Kedah and Perak has resulted in the discovery of 56 orchid species from 37 genera. From this account, 12 species are new records within Kedah and 2 species are new records within Malaysia, namely *Bulbophyllum meson* J.J.Verm., Schuit. & de Vogel and *Luisia brachystachys* (Lindl.) Blume. Three species are endemic to Peninsular Malaysia which are *Anoectochilus sanguineus* P.T.Ong & P.O'Byrne, *Cheirostylis goldschmidtiana* Schltr. and *Phalaenopsis appendiculata* Carr. Findings of two keystone species, the long-lost *Cheirostylis goldschmidtiana* and the endangered snow-white slipper orchid, *Paphiopedilum niveum* (Rchb.f.) Stein, have catalyzed the need for a more comprehensive study to unveil the species richness and endemism within a limestone habitat. Heat maps using geographical data produced from ArcGIS software have enabled precise determination of the areas with the highest concentration of orchid diversity. Results from this study have shown that Gunung Fakir Terbang houses the greatest number of orchid species, followed by Gunung Batu Putih and Gunung Baling. Meanwhile, lower species occurrences were recorded from Gunung Pulai and Gunung Pong. However, lack of attention and delayed conservation action on this unique karst habitat would only lead to more habitat loss, leading to the decline of limestone orchid populations. Orchids as well as other plants are threatened by anthropogenic activity such as quarrying and forest clearing for agriculture. Thus, the results of this study will serve as baseline data for future work in documenting and conserving limestone orchids and their karst habitat in northern Peninsular Malaysia.

**Keywords:** Orchidaceae; diversity; limestone; heat map; Peninsular Malaysia; conservation





## 1. Introduction

The state of Kedah lies in the northern part of Peninsular Malaysia, covering a total area of 9425 km$^2$, bordered to the north by Perlis and Thailand, Pulau Pinang to the southwest and Perak to the south. There is over 330,585 ha of forested area throughout Kedah [1]. Two mountain ranges occupy some parts of Kedah, which are Banjaran Kedah–Singgora in the north and Banjaran Bintang in the east. Banjaran Kedah–Singgora has been the focus area for many geological studies, especially the limestone hills [2–4].

Limestone hills were reported throughout Kedah to Perlis, the Kinta Valley in central part of Perak, the Klang Valley between Selangor and Kuala Lumpur, Kelantan focusing



in the Gua Musang area, and Pahang [3]. A total of 173 limestone outcrops have been mapped in Kedah alone [4], and yet more are to be included in the list, particularly in the small, isolated karst complexes in remote areas with limited access. In Kedah, most of the limestone hills are scattered throughout the north, bordering Perlis. Several prominent hills are located towards the south of Kedah, concentrated in the district of Baling, with a few located within the border of Perak in Pengkalan Hulu and Gerik. These hills, such as Gunung Baling and Gunung Pulai, have recorded numbers of rare and endemic species such as the endangered snow-white slipper orchid, *Paphiopedilum niveum* (Rchb.f.) Stein and the long–lost *Cheirostylis goldschmidtiana* Schltr. The discovery of these species has emphasized the need for thorough documentation on the diversity of orchids in Kedah's limestone hills to prevent their extinction from various anthropogenic activities.

This karst habitat also has proven to be a hotspot for high levels of species endemicity. The inaccessibility due to the unique yet rough topography with highly variable climatic and edaphic conditions has resulted in high diversity and endemism on limestone formations [5–7]. In Peninsular Malaysia, about 21% of the plant species present in karst habitats are endemic species [2]. However, the status of our limestone hills has become a concerning issue, as they are not gazetted as protected areas, even though this unique habitat is home to hundreds endemic species of flora and fauna.

Various botanical surveys on karst habitats have been carried out in previous years. However, these botanical excursions were possible only in the more accessible parts of the north and west limestone hills. Records on past botanical surveys of the limestone hills in the northern part of Kedah are focused only on better–known localities such as in the district of Baling and Sik. Besides having relatively low collection levels in this area, no thorough documentation has been published which enhances the limited knowledge on the diversity of limestone orchids in Kedah. The only prominent reference available is a checklist prepared for the limestone orchids along the Nakawan Range in Perlis [8].

Due to the scarcity of main references and the lack of data, diversity assessments on the limestone orchids in the northern part of Peninsular Malaysia should not be further delayed. Despite being rich in biodiversity, few botanical observations have focused primarily on the limestone flora. Specifically, no updated checklist on orchids within a karst habitat in Kedah and Perak has yet been reported. Diversity assessments on the orchids of Kedah, particularly on limestone hills, are insufficiently studied, and records on endemic and rare species are very limited due to the harsh topography responsible for the area's inaccessibility.

Lack of attention and delayed conservation action can only result in further habitat loss, which could lead to the extirpation of limestone orchids endemic to this unique habitat. Obscure species which may be previously unknown to science are threatened by quarrying activity and forest clearing for agriculture. Reports on change in forest coverage in Kedah have shown a significant decrease in the total cover of forested area by 31,583 ha, due to land use activities such as land clearing for the development of oil palm plantations, access roads, housing, and water bodies [1]. Within a period of 29 years, deforestation for various purposes has led to a decrease of about 9% in forest cover between 1988 and 2017. Wild orchids are being exposed towards logging especially epiphytic orchids, as most of their habitats are in a single emergent tree [9,10].

The loss of wild orchid species in limestone forests is also attributable to destructive human activity such as that of orchid collectors who harvest from the wild. The surge in illegal wild orchid collection is very concerning with regard preserving these rare and endangered orchids. Wild orchids have been smuggled out from their natural habitat to be sold to avid orchid collectors and amateurs [8]. Wild orchids have always been in demand due to their unique flower and exotic nature.

This study aims to evaluate species richness and endemism of the limestone orchid flora from the karst habitat in northern Peninsular Malaysia, by concentrating on the states of Kedah and Perak. An updated species list will be prepared, and specific limestone hills which house the greatest number of orchid species will be determined as hotspot areas. Among the main objectives of this study is the identification of conservation priorities of

any limestone orchid that is under serious threat of extinction. The results of this study also will serve as an important reference to move further towards conservation initiatives for endangered limestone orchid species and their unique karst habitat.

## 2. Materials and Methods

### 2.1. Study Area

A total of 12 field assessments were carried out from 2021 to 2022 in five limestone hills along the Kedah–Singgora Range, as shown in Figure 1. The karst habitats in focus are Gunung Fakir Terbang that lies within the Ulu Muda Forest Reserve (bordering to Thailand to the north), Gunung Batu Putih and Gunung Pong (bordering Gerik–Pengkalan Hulu, Perak), and Gunung Baling and Gunung Pulai which are located in the central district of Baling. These five limestone hills were selected based on their current state of vulnerability and historical botanical records.

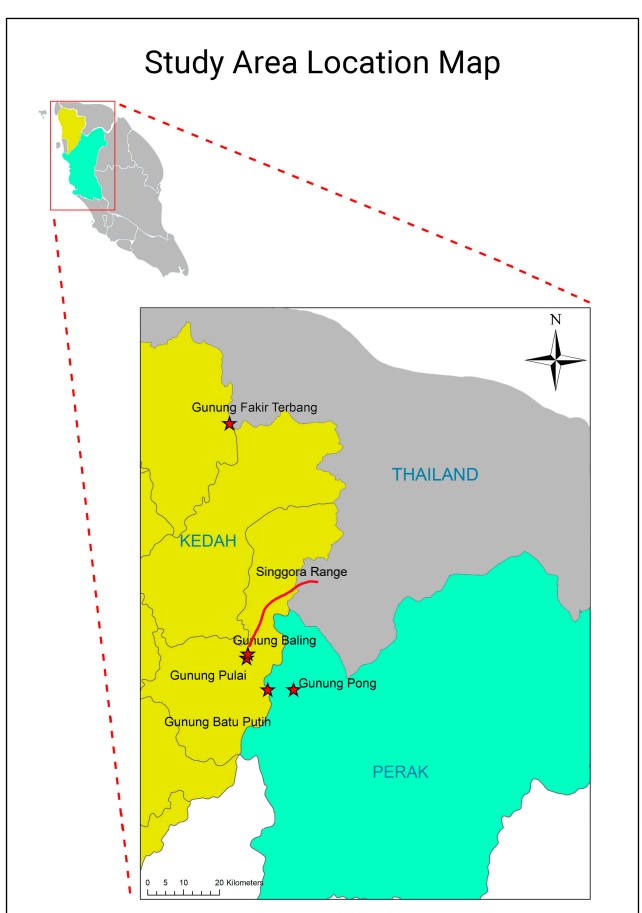

**Figure 1.** Map of Kedah and Perak, showing the study sites along the Singgora Range for limestone orchid diversity assessments.

### 2.2. Data Collections and Identification

The collections were made based on convenience sampling method along the forest trails from base of the foothill ascent to the peak of all five limestone hills and vice versa. Species identifications were conducted in the field based on their macromorphological characteristics. For discreet micromorphological diagnostics of selected orchid species, specimens were further examined under field and stereo microscopes. The specimens were identified using the morphological characteristics described, and the identification keys prepared by preceding authors [8,11–15]. The current accepted names for species were validated through the updated online database—Plants of the World Online (POWO) [16]—and checklists made available regarding Peninsular Malaysia's orchids [15,17].

In order to comply with the species and habitat preservation and conservation practices, most of the plant specimens were documented in the form of photographs, and collection of living specimens was to be circumvented in any way possible. Preserved herbarium specimens and spirit materials of the samples collected were deposited in the herbarium at the School of Biological Sciences, Universiti Sains Malaysia (USMP) [18]. Living collections of sterile specimens for further identification, ex situ conservation purposes and germplasm studies were cultivated in the orchidarium, School of Biological Sciences, Universiti Sains Malaysia.

### 2.3. Sampling Density Analysis

Geographical coordinates were primarily obtained directly in each physical sighting of the orchids during field assessments. The species occurrences in each population were then georeferenced using mapping software (ArcGIS, ArcMap Version 10.3), projecting 102 geocoordinates to form a heat map. This software uses kernel density estimation (KDE) as the type of spatial analysis to identify spatial clusters of high- and low-value hot spots.

## 3. Results

### 3.1. Species Composition

A total of 56 species of orchids from 37 genera were identified from all five limestone hills. From the account, 44 species are epiphytic orchids, 7 are terrestrials and 5 species are true lithophytes. An outstanding finding from this research is the discovery of two orchid species which are new records within Malaysia, namely *Bulbophyllum meson* J.J.Verm., Schuit. and de Vogel and *Luisia brachystachys* (Lindl.) Blume. Meanwhile, 12 species are recognized as new records in Kedah which are hitherto unlisted earlier publications. The five lithophytic orchids which can only be found growing within a limestone habitat are *Aerides krabiensis* Seidenf., *Cheirostylis goldschmidtiana* Schltr., the rare *Spathoglottis hardingiana* C.S.P.Parish and Rchb.f., the endangered *Paphiopedilum niveum* (Rchb.f.) Stein and the outstanding *Vandopsis gigantea* (Lindl.) Pfitzer. These five species also have a very narrow distribution area, occurring only in the limestone hills in the north of Peninsular Malaysia, and stretching upwards to the southern tip of Thailand.

Three species of orchids, *Anoectochilus sanguineus* P.T.Ong and P.O'Byrne, *Cheirostylis goldschmidtiana* Schltr. and *Phalaenopsis appendiculata* Carr which are endemic to Peninsular Malaysia were also discovered in this study. Table 1 shows the list of orchids recorded from all five limestone hills and their current conservation status. Specimen comparisons were made by examining voucher specimens (physically and virtually) from various collectors, and holdings of several herbaria such as KEP, KLU, SAN, SING, K and L. The current conservation status for each orchid species was assessed through the IUCN Red List of Threatened Species online database (Version 2022–2) [19]. Majority of the species is yet to be assessed and was assigned into the category of Not Evaluated (NE). The common and widespread species, *Coelogyne foerstermannii* and *Dendrobium aloifolium*, are categorized as Least Concern (LC), while the heavily threatened *Paphiopedilum niveum* is listed as Endangered (EN).

**Table 1.** Checklist of the orchids found on all five limestone hills in Northern Peninsular Malaysia from this study, with comparisons made to previous records. The voucher specimens were deposited in USMP.

| | Species | Distribution in Peninsular Malaysia | Collection Number/Herbarium Number | Collector | Conservation Status |
|---|---|---|---|---|---|
| 1. | *Acriopsis liliifolia* (J.Koenig) Ormerod | Widespread | FRI67602 SNB047/USMP11890 | Ong P.T. Shahrul N. | NE |
| 2. | *Aerides odorata* Lour. | Widespread | FRI67680 KLU30599 SNB037/USMP11891 | Kiew R. Liew K.C. Shahrul N. | NE |
| 3. | *Aerides krabiensis* Seidenf. | Gunung Baling, Kedah | SNB062/USMP11892 | Shahrul N. | NE |
| 4. | *Ania penangiana* (Hook.f.) Summerh. | Kedah, Perak, Penang | FRI67685 SNB024/USMP11893 | Ong P.T. Shahrul N. | NE |

**Table 1.** *Cont.*

| | Species | Distribution in Peninsular Malaysia | Collection Number/Herbarium Number | Collector | Conservation Status |
|---|---|---|---|---|---|
| 5. | ***Anoectochilus sanguineus* P.T.Ong andand P.O'Byrne** * | Kedah, Perak | FRI71388 SNB066/USMP11894 | Ong P.T. andand O'Byrne P. Shahrul N. | NE |
| 6. | *Agrostophyllum majus* Hook.f. | Widespread | FRI64199 SNB026/USMP11895 | Ong P.T. Shahrul N. | NE |
| 7. | *Apostasia nuda* R.Br. | Widespread | FRI80771 KLU9382 SNB051/USMP11896 | Ong P.T. Stone B.C. Shahrul N. | NE |
| 8. | *Arachnis flos–aeris* Rchb.f. * | Perak, Pahang Selangor, Negeri Sembilan | FRI71127 KLU17310 SNB028/USMP11897 | Pauzi H. Ying Y.M. Shahrul N. | NE |
| 9. | *Bulbophyllum lilacinum* Ridl. | Pulau Langkawi, Gunung Jerai, Kedah | FRI67702 SNB042/USMP11898 | Ong P.T. Shahrul N. | NE |
| 10. | *Bulbophyllum limbatum* Lindl. * | Johor, Singapore | FRI71318 SNB054/USMP11899 | Ong P.T. Shahrul N. | NE |
| 11. | *Bulbophyllum medusae* (Lindl.) Rchb.f. | Widespread | FRI75397 SNB044/USMP11900 | Ong P.T. Shahrul N. | NE |
| 12. | *Bulbophyllum meson* J.J.Verm., Schuit. and de Vogel ** | Gunung Fakir Terbang, Kedah | K000891022 SNB052/USMP11901 | Kerr, A.F.G. Shahrul N. | NE |
| 13. | *Bulbophyllum planibulbe* (Ridl.) Ridl. | Kedah, Pahang | FRI79841 SNB057/USMP11902 | Yap J.W. Shahrul N. | NE |
| 14. | *Bulbophyllum purpurascens* Teijsm. and Binn. | Widespread | FRI88603 SNB048/USMP11903 | Ong P.T. Shahrul N. | NE |
| 15. | ***Cheirostylis goldschmidtiana* Schltr.** * | Gunung Baling, Gunung Pong, Kedah | SNB006/USMP11904 | Shahrul N. | NE |
| 16. | *Cleisostoma* sp. | Gunung Baling, Kedah | SNB010/USMP11905 | Shahrul N. | NE |
| 17. | *Cleisomeria lanata* (Lindl.) Lindl. ex G.Don | Kedah, Pahang, Johor | L.1500004 SNB050/USMP11906 | Larsen et al. Shahrul N. | NE |
| 18. | *Coelogyne asperata* Lindl. | Widespread | FRI67494 SNB029/USMP11907 | Ong P.T. Shahrul N. | NE |
| 19. | *Coelogyne foerstermannii* Rchb.f. | Widespread | SAN120788 SNB033/USMP11908 | Joseph et al. Shahrul N. | LC |
| 20. | *Coelogyne trinervis* Lindl. | Pulau Langkawi, Kuala Lumpur, Penang, Pahang | FRI89830 SNB049/USMP11944 | Yap J.W. Shahrul N. | NE |
| 21. | *Corymborkis veratrifolia* (Reinw.) Blume | Widespread | FRI64084 KLU48160 SNB015/USMP11945 | Ong P.T. Zulkapli et al. Shahrul N. | NE |
| 22. | *Crepidium prasinum* (Ridl.) Szlach. * | Perlis | KLU1712 SNB021/USMP11946 | Mohd Kasim Shahrul N. | NE |
| 23. | *Cymbidium atropurpureum* (Lindl.) Rolfe * | Pahang, Melaka, Johor | FRI75468 KLU30235 SNB020/USMP11947 | Ong P.T. Dransfield J. Shahrul N. | NE |
| 24. | *Cymbidium haematodes* Lindl. * | Kedah, Penang | FRI67593 SNB064/USMP11948 | Ong P.T. Shahrul N. | NE |
| 25. | *Dendrobium aloifolium* (Blume) Rchb.f. | Widespread | FRI86220 KLU27902 SNB018/USMP11949 | Yap J.W. Chin S.W. Shahrul N. | LC |
| 26. | *Dendrobium crumenatum* Sw. | Widespread | FRI91022 KLU27487 SNB058/USMP11950 | Siti Munirah M.Y. Stone B.C. Shahrul N. | NE |
| 27. | *Dendrobium farmeri* Paxton | North of P. Malaysia | FRI75581 SNB040/USMP11951 | Ong P.T. Shahrul N. | NE |
| 28. | *Dendrobium indragiriense* Schltr. | Widespread | FRI67469 SNB060/USMP11952 | Ong P.T. Shahrul N. | NE |
| 29. | *Dendrobium leonis* (Lindl.) Rchb.f. | Widespread | FRI89811 SNB041/USMP11953 | Yap J.W. Shahrul N. | NE |
| 30. | *Dendrobium linguella* Rchb.f. | Widespread | FRI70253 KLU9376 SNB053/USMP11954 | Chan M.Y. Stone B.C. Shahrul N. | NE |

**Table 1.** *Cont.*

| | Species | Distribution in Peninsular Malaysia | Collection Number/Herbarium Number | Collector | Conservation Status |
|---|---|---|---|---|---|
| 31. | *Dendrobium pachyphyllum* (Kuntze) Bakh.f. | Widespread | FRI71120 SNB013/USMP11955 | Ong P.T. Shahrul N. | NE |
| 32. | *Dendrobium plicatile* Lindl. | Widespread | FRI87109 KLU49436 SNB008/USMP11956 | Imin K. et al. Yong K.T. Shahrul N. | NE |
| 33. | *Dendrolirium ornatum* Blume | Extreme north | FRI44556 SNB030/USMP11957 | Keith H.G. Shahrul N. | NE |
| 34. | *Eria javanica* (Sw.) Blume | Widespread | FRI80934 SNB005/USMP11958 | Ong P.T. Shahrul N. | NE |
| 35. | *Grosourdya appendiculata* (Blume) Rchb.f. | Widespread | FRI59596 SNB065/USMP11959 | Jutta M. Shahrul N. | NE |
| 36. | *Habenaria reflexa* Blume * | Perak, Pahang | FRI88524 SNB014/USMP11960 | Ong et al. Shahrul N. | NE |
| 37. | *Luisia brachystachys* (Lindl.) Blume ** | Kedah | K000891542 SNB055/USMP11961 | Wallich, N. Shahrul N. | NE |
| 38. | *Luisia zollingeri* Rchb.f. | Perlis, Pulau Langkawi | K000891537 SNB059/USMP11962 | Kerr, A.F.G. Shahrul N. | NE |
| 39. | *Nervilia concolor* (Blume) Schltr. | Penang, Perak, Johor | FRI80634 KLU40705 SNB001/USMP11963 | Ong et al. Raulerson L. Shahrul N. | NE |
| 40. | *Oberonia* sp. | Kedah | SNB038/USMP1164 | Shahrul N. | NE |
| 41. | *Oxystophyllum carnosum* Blume | Widespread | L1498123 SNB067/USMP11965 | Soinin et al. Shahrul N. | NE |
| 42. | *Paphiopedilum niveum* (Rchb.f.) Stein | Perlis, Kedah, Kelantan | FRI56091 SNB031/USMP11966 | Julius A. and Angan A. Shahrul N. | EN |
| 43. | *Phaius amboinensis* Blume | Kedah, Perak, Johor | K000891005 SNB055/USMP11967 | Parish, E.C. Shahrul N. | NE |
| 44. | ***Phalaenopsis appendiculata* Carr *** | Pahang | K000891360 SNB019/USMP11968 | Henderson M.R. Shahrul N. | NE |
| 45. | *Phalaenopsis fuscata* Rchb.f. * | Pahang | FRI59591 SNB068/USMP11969 | Jutta M. and Kueh H.L. Shahrul N. | NE |
| 46. | *Pholidota imbricata* Hook. | Widespread | FRI60377 KLU20163 SNB027/USMP11970 | Yong et al. Chung C.S. Shahrul N. | NE |
| 47. | *Pinalia floribunda* (Lindl.) Kuntze | Widespread | K001127303 SNB043/USMP11971 | Prince J. Shahrul N. | NE |
| 48. | *Pomatocalpa spicatum* Breda * | Perak, Pahang, Negeri Sembilan | FRI53222 KLU29757 SNB016/USMP11972 | Phoon et al. Chin S.C. and Chia L.T. Shahrul N. | NE |
| 49. | *Renanthera elongata* (Blume) Lindl. | Widespread | KLU44409 SNB009/USMP11973 | Chan C.L. Shahrul N. | NE |
| 50. | *Renanthera histrionica* Rchb.f. | Widespread | FRI60358 KLU20028 SNB046/USMP11974 | Yong W.S.Y. and Vermeulen J.J. Chin S.C. Shahrul N. | NE |
| 51. | *Sarcoglyphis comberi* (J.J.Wood) J.J.Wood * | Widespread | K000942457 SNB069/USMP11975 | Comber J.B. Shahrul N. | NE |
| 52. | *Spathoglottis hardingiana* C.S.P.Parish and Rchb.f. | Pulau Langkawi | KLU29694 USMP12212 SNB012/USMP11976 | Chin S.C. and Chia L.T. Farah Alia N. Shahrul N. | NE |
| 53. | *Taeniophyllum pusillum* (Willd.) Seidenf. and Ormerod | Widespread | FRI64176 SNB061/USMP11977 | Ong P.T. Shahrul N. | NE |
| 54. | *Trichotosia ferox* Blume | Widespread | FRI41617 SNB045/USMP11978 | Perumal et al. Shahrul N. | NE |
| 55. | *Tropidia angulosa* (Lindl.) Blume | Widespread | FRI75500 SNB003/USMP11979 | Ong et al. Shahrul N. | NE |
| 56. | *Vandopsis gigantea* (Lindl.) Pfitzer | Pulau Langkawi, Terengganu, Melaka | BM000538893 SNB004/USMP11980 | Ridley N. H. Shahrul N. | NE |

*Note.* Endemic species are mentioned in bold text; * New record to Kedah; ** New record to Malaysia.

### 3.2. Heat Map Distributional Data on Species Occurrences

Figure 2A–E display the 102 geocoordinates of species occurrences (individuals of the 56 species) on all five limestone hills. The heat maps produced show varying results using the species occurrences (number of species and total individuals) as the weighted score with KDE. It starts by overlaying a grid over the research region and estimating density based on the centers of each grid cell [20]. On the heat maps, several clusters were formed showing the densest spots representing high species richness throughout all five study sites. Gunung Fakir Terbang (Figure 2E) has the highest occurrences of orchid species with 17 species and 10 genera, followed by Gunung Batu Putih with 19 species and 16 genera (Figure 2C) and Gunung Baling with 13 species and 11 genera (Figure 2A). Smaller clusters were observed in Gunung Pulai with 10 species and 10 genera (Figure 2B), while Gunung Pong (Figure 2D) has 14 species and 12 genera, concentrating on one spot with high species occurrences.

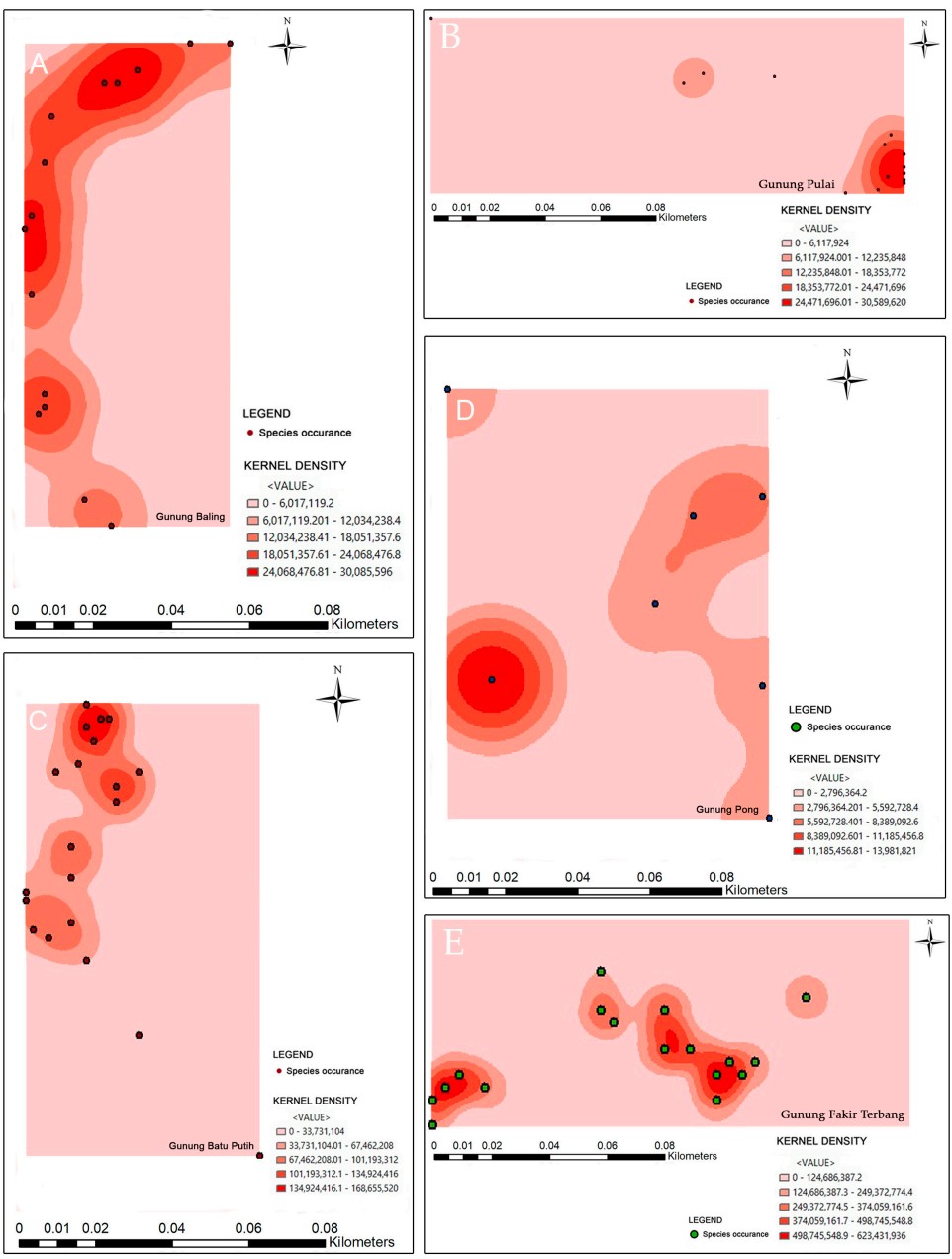

**Figure 2.** Heat maps showing the results of kernel density estimation of orchid observations in (**A**) Gunung Baling, (**B**) Gunung Pulai, (**C**) Gunung Batu Putih, (**D**) Gunung Pong, (**E**) Gunung Fakir Terbang. Hotspots graded from pinkish to dark red showing increasing observational findings.

## 4. Discussion

Endemism usually occurs through isolated or restricted areas in which possible survivors within certain species from the primitive stock are subjected to a catastrophic geological event [21,22]. Although deforestation has impacted the structure of forested areas over the years, it has also provided easier access to the remote and previously inaccessible parts of the forest. Due to being secluded from others, these previously inaccessible regions have been proven to contain a high number of new species and newly recorded species for that area [23], which explains the discovery of newly recorded orchids in this study.

Figure 3A–D shows four out of all 14 newly recorded orchid species discovered in this study. The miniature orchid *Bulbophyllum meson* (Figure 3A), was identified in Ulu Muda Forest Reserve, growing as an epiphytic. The small, carunculate bulbs were embedded strongly on the rough and cracked trunk of a tree. This species was first discovered and described by Gunnar Seidenfaden and Tem Smitinand as *Trias intermedia* in 1965. However, taxonomic revision supported by morphological examinations and DNA evidence has generated new combinations in *Bulbophyllum*, formally transferring *Trias*, together with several other satellite genera, to *Bulbophyllum* [24]. This transfer was made due to the similarities of vegetative structures and flower attributes shared by species among the two genera. Phylogenetic analysis using two plastid genes, *rbcL* and *matK*, and one nuclear region nrITS, has shown that all species of *Bulbophyllum* in section *Trias* were embedded among other groups of *Bulbophyllum* [25]. *Bulbophyllum meson* was previously known only in Thailand; however, it was discovered during the visit to Gunung Fakir Terbang in the Ulu Muda Forest Reserve, making it a newly recorded species within Malaysia. Another new record within Malaysia, discovered from the same location, was *Luisia brachystachys* (Figure 3B). *Luisia brachystachys* is often confused with *L. zollingeri*, as both species show a high degree of resemblance in vegetative characteristics and in the shape of the labellum. The two species possess an almost spear–shaped epichile, but where *L. brachystachys* differs from *L. zollingeri* is in the presence of indistinct, shallow ridges between the hypochile and epichile. The discovery of these two species as new records within Malaysia somehow was expected, as the eastern side of Gunung Fakir Terbang lies within Thailand. This is in agreement with Ridley [26], who noted that Thai flora may extended further down towards the north part of Peninsular Malaysia.

Another fascinating finding from this study was the re–collection of *Cheirostylis goldschmidtiana* outside its type location (Figure 3C). It was first described in 1915 from a specimen from Penang Hill, which is the only known locality for *C. goldschmidtiana*. This narrowly endemic species had not been sighted again since its first discovery, probably due its die–back nature or the lack of further botanical excursions to its locality type. In 2010, Rogier van Vugt recorded the same species from Baling, and photographs of it were published online through the website of the Swiss Orchid Foundation of the Herbarium Jany Renz. *Cheirostylis goldschmidtiana* was sighted again in this study in several locations, and it can thus be concluded that the native range of this rare terrestrial orchid is northern Peninsular Malaysia.

Meanwhile, *Habenaria reflexa* (Figure 3D), with the flowers resembling flying mosquitoes, is a new record within Kedah. This species was only previously known from Perak and Pahang in the east coast of the peninsula.

*Spathoglottis hardingiana*, which was previously thought to be confined to the limestone vegetation in Pulau Langkawi, was also discovered in this study. Interestingly, the size of the plants sighted in the mainland was larger—almost double to triple the size of plants described in Pulau Langkawi. The limestone outcrops in the Baling–Pengkalan Hulu complex (the Kedah–Singgora Range) share the same geological attributes as those in Pulau Langkawi (Machinchang Formation); both arose at almost the same time, more than ~500 million years ago. The orchid flora of Pulau Langkawi also show close proximity to those of Thailand; thus, it is somewhat expected to find almost the same species composition in the limestone hills of Baling–Pengkalan Hulu and Perlis. Figure 4A–B shows the habit and flowers of *S. hardingiana* in its natural karst habitat in Baling.

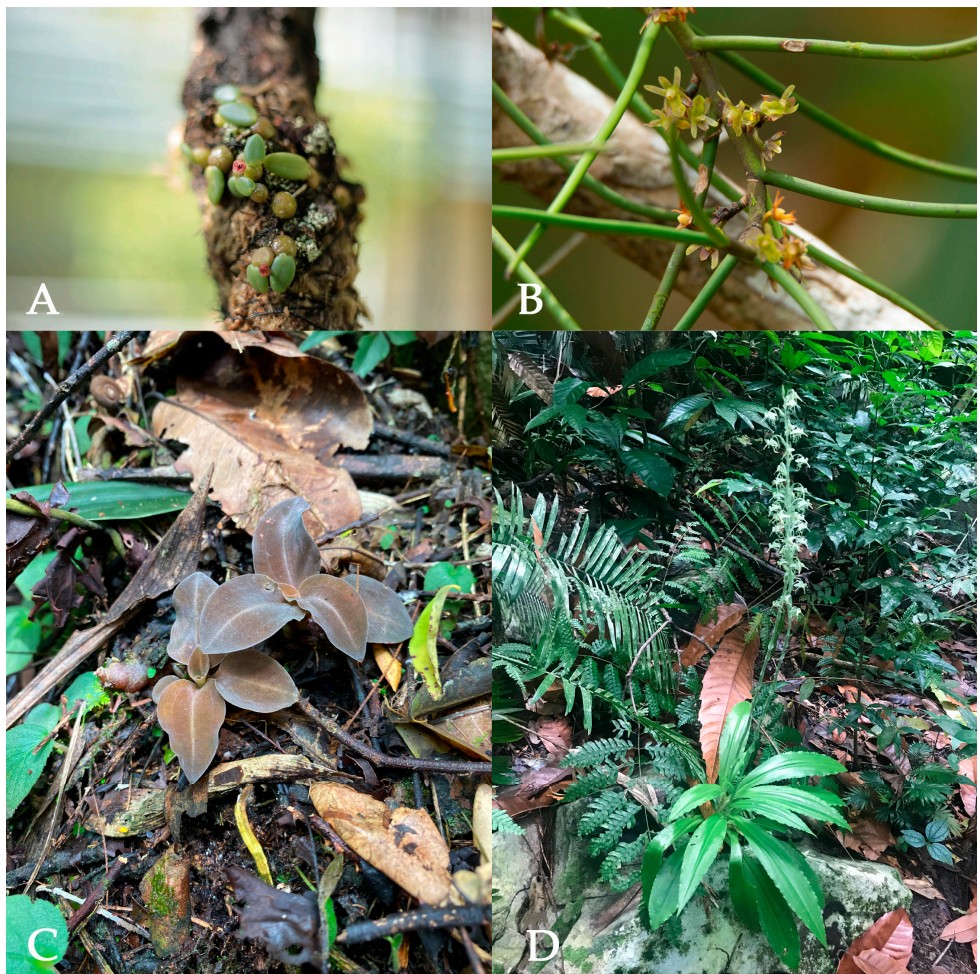

**Figure 3.** Newly recorded orchid species within Malaysia: (**A**) *Bulbophyllum meson*; (**B**) *Luisia brachystachys*. Newly recorded orchid species within Kedah (**C**) *Cheirostylis goldschmidtiana*; (**D**) *Habenaria reflexa*. Photos by Shahrul Nizam Abu Bakar and Mohd Mushahril Abdul Shukor.

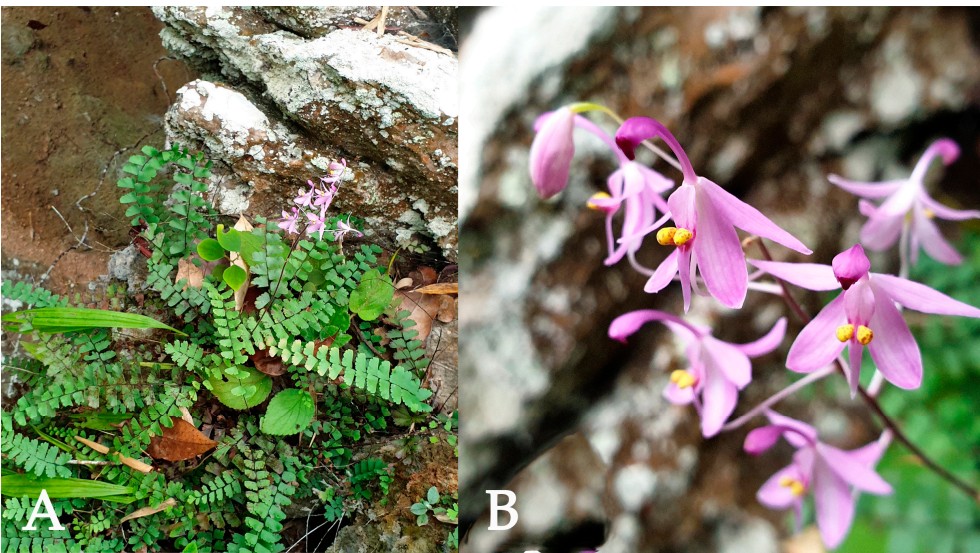

**Figure 4.** *Spathoglottis hardingiana* in its natural habitat: (**A**) habitat; (**B**) flowers. Photos by Muhamad Faizal Md Azmi.

The large-scale wild orchid commercialization by locals and the illegal collection of wild orchids have caused major threats to the species. For example, the population size of the white slipper orchid, *Paphiopedilum niveum* (Figure 5A,B), which is endemic to the limestone areas, shows a worrying declining trend [27]. The same condition was also observed in other wild *Paphiopedilum* species such as *P. barbatum* in Johor and *P. callosum* var. *sublaeve* from Gunung Jerai. The admirably named "Lady's Slipper" orchid is known for its variety in the shape, size and color of the flower, which makes it among the most in-demand [28]. The genus compromises about 80 species which occur throughout Southeast Asia, and many of them are narrowly endemic species. As for *P. niveum*, the native range of this species is restricted to the extreme south of Peninsular Thailand towards northern Peninsular Malaysia. This rare species has been validated as an endangered species, listed in the IUCN Red List [19,29]. Throughout the 12 months of field assessments and the long-term observations made by the corresponding author since 2014, the population size of *P. niveum* in all five limestone hills is deteriorating. This is undoubtedly the result of human disturbances, mainly by ruthless collection for commercial trade. Since the price in local markets is often low, wild orchids, including this endangered species, are often collected in bulk.

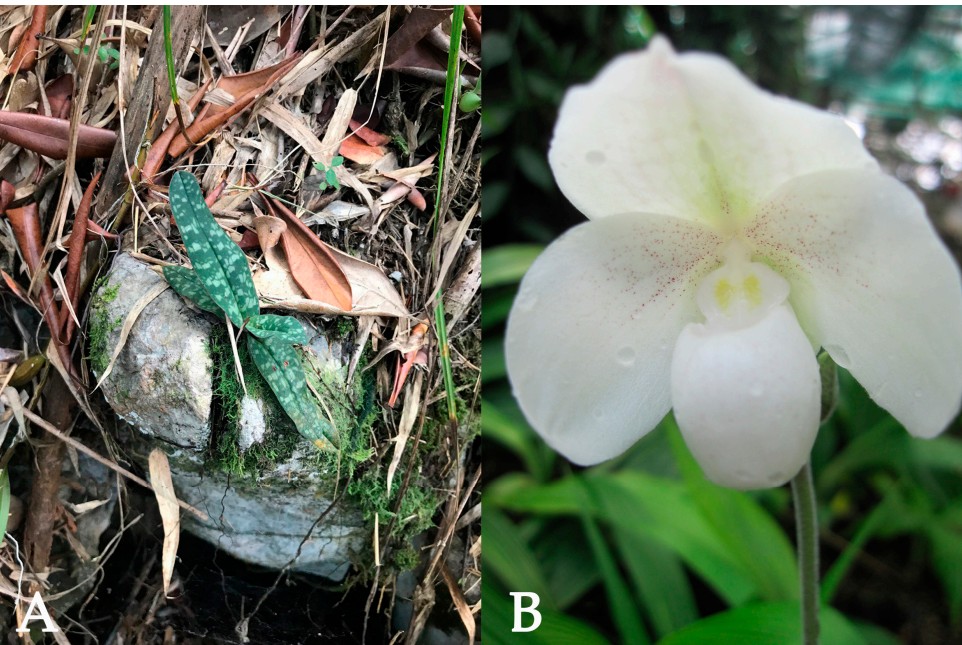

**Figure 5.** Population of *Paphiopedilum niveum* in Gunung Baling and Gunung Pong: (**A**) habitat; (**B**) flowers. Photos by Farah Alia Nordin.

Based on the heat maps produced using the ArcGIS software, Gunung Fakir Terbang, Gunung Batu Putih and Gunung Baling have been recognized to have the highest concentration of wild orchids (in terms of species numbers and individuals) compared to Gunung Pulai and Gunung Pong. The three limestone hills were thus identified as hot spot areas with prioritized conservation concerns. Gunung Fakir Terbang and Gunung Baling, for example, are threatened by disturbances such as land clearing for certain logging and quarrying activity within their vicinity. The utilization of ArcGIS in determining hot spot areas that are rich in species diversity or under irreversible environmental threats will speed up the recognition of areas that require immediate conservation action.

The wild orchid populations in Peninsular Malaysia also facing serious threats due to the permanent loss of habitat or habitat degradation caused by deforestation and anthropogenic activity. These threats are strongly affecting the population of orchids by disturbing their complex ecological interactions, such as epiphytic growing on host trees, interactions with pollinators and obligatory relationships with the soil mycorrhizal fungi. The wild

orchids are thus exposed to a higher risk, as they are dependent on other organisms that are also being affected by habitat loss or climate change. Figure 6A–D shows selective logging activities in one of the study sites.

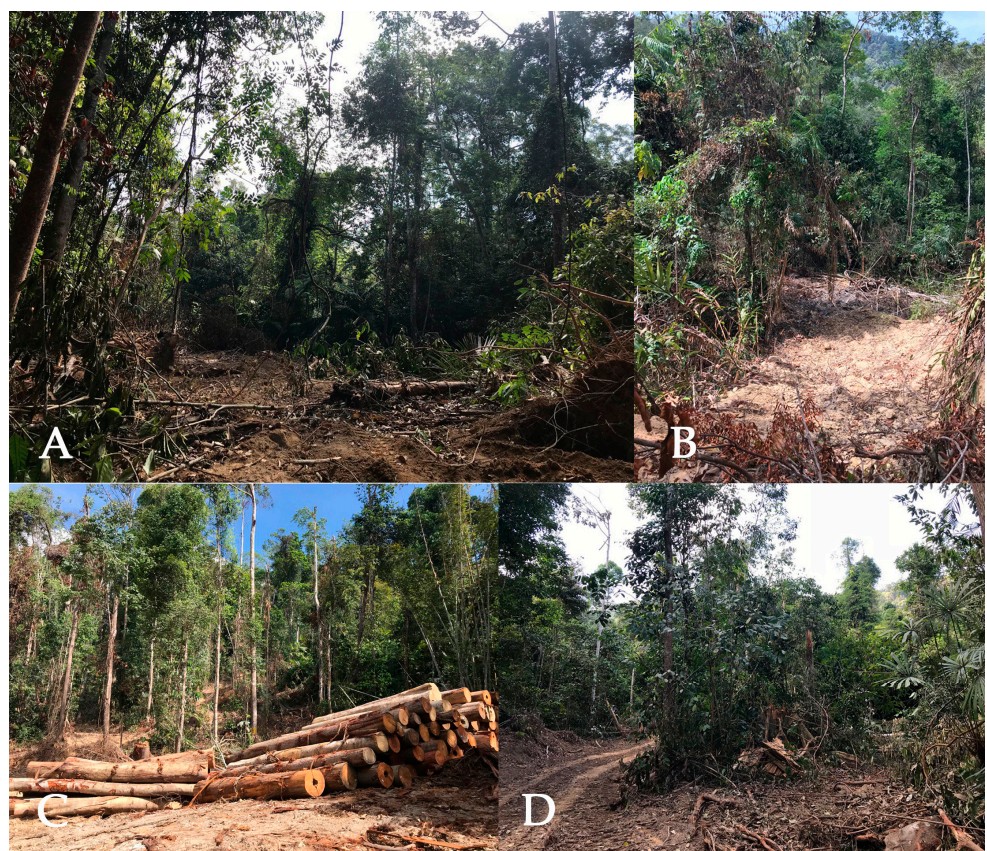

**Figure 6.** Forest clearing through logging in several locations within the study areas: (**A**,**B**) disturbed areas due to heavy logging machinery; (**C**) sites for sawn-off timber; (**D**) cleared trails for operations and extraction of timber. Photos by Shahrul Nizam Abu Bakar.

The survival of wild orchids in an active logging site is compromised by the disturbed interactions between the orchids and contributing organisms. For example, once the forests have been cleared, the climatic conditions of that area will change. The changes in levels and quality of light, and prevailing wind patterns may influence the distribution and survival of orchids [30,31]. Alarmingly, the two existing quarry sites in Gunung Baling may result in changed habitat conditions, due to quarrying operations which include vegetative clearing and earthworks to prepare the site for production areas, rock blasting and extraction.

Thus, immediate conservation action needs to be materialized promptly. Strict guidelines need to be enforced to mitigate the loss of diversity in this unique habitat. Wild orchid preservation and conservation are key to preserving and securing the rich and wildly captivating orchid legacy we have today for future generations.

## 5. Conclusions

The limestone hills in northern Peninsular Malaysia harbor a great diversity of wild orchids, with a total of 56 species observed here. Two species are new records within Malaysia, 12 species are new records within Kedah and three are endemic to Peninsular Malaysia. Further diversity assessments on the karst habitat are highly recommended, as more species are yet to be documented from the other minor limestone hills which remain unexplored. Most importantly, immediate conservation action needs to be embarked on to protect the diversity and endemism of wild orchid species which this unique habitat holds.

**Author Contributions:** Data curation, S.N.A.B., F.A.N., F.R., M.H.J., M.F.M.A. and R.Z.; formal analysis and investigation, S.N.A.B., F.A.N., M.F.M.A. and A.A.R.; methodology, F.A.N., A.A.R., A.S.O., A.R. and R.Z.; resources, F.A.N., A.S.O., A.R. and A.A.R.; supervision, F.A.N., A.A.R. and R.Z.; validation, F.A.N. and A.A.R.; writing—original draft, S.N.A.B.; writing—review and editing, S.N.A.B., F.A.N. and A.R. All authors have read and agreed to the published version of the manuscript.

**Funding:** The study was funded by The American Orchid Society Conservation Grant (C2020–17) of assignment No. 304/PBIOLOGI/6501097/A151 and National Conservation Trust Fund for Natural Resources (Ministry of Natural Resources, Environment and Climate Change Malaysia) of assignment No. 304/PBIOLOGI/6501324/K130 awarded to the corresponding author. All funders provided financial supports for the present work but did not contribute any additional role in the research design, data collections and analysis, and preparation of the manuscript.

**Data Availability Statement:** Data presented in this article are available on request from the corresponding author.

**Acknowledgments:** The authors would like to express our deepest gratitude to the administrative and field personnel of School of Biological Sciences, Universiti Sains Malaysia and Baling District Council for the facilities and assistance provided during the study conducted. Thank you to Forestry Department of Peninsular Malaysia and Kedah State Forestry Department for the research permits granted with number KT7767–7775/2021. Many thanks to Richard Chung (KEP), Ong Poh Teck (KEP) and Yong Kien Thai (KLU) for the assistance provided during visits to the herbaria. Sincere thanks to Tom Miranda from The American Orchid Society (AOS) for his valuable supports towards this conservation study. Special thanks to Muhammad Azlan Md Azmi, Mohd Asri Md Azmi, Syed Mohd Edzham Syed Hamzah, Umi Natra Shaharudin and Mohd Zulhusni Mohammad Roslan for their immense contribution in the fields.

**Conflicts of Interest:** The authors declare no conflict of interest.

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
