# Peer review of "Unveiling Limestone Orchid Hotspots in the Karst Hills of Northern Peninsular Malaysia"

_diversity, doi:10.3390/d15070819_

Round 1
Reviewer 1 Report
Bakar et al. carried out a series of 12 months diversity assessments on five limestone hills in Kedah and Perak, and recorded 56 orchid species from 37 genera. Undoubtedly, this work filled the gap on the diversity of orchids from the limestone hills in the northern part of Peninsular Malaysia, particularly in the states of Kedah and Perak. Thus, this ms is suitable for formally published in Diversity. The data is reliable and the organization of paper is well, so it could be published before following minor changes:
1) In this study, the authors reported they recorded 56 orchids species. However, in lines 172-173, they mentioned “Figure 2A–E display the 102 species occurrences (geocoordinates) on all five limestone hills”. Here, where the 102 species come from? What relationship between 56 orchids species and 102 species?
2) In 3.2. Heat map distributional data, if the authors can use the data of narrow distributed orchid species to establish a heat map, it will give more evidence to identify the key areas for conservation;
3) Lines 291-293, the authors mentioned “The changes in levels and quality of light, and prevailing wind patterns may influence the distribution and survival of orchids [28,29]”. However, I have not found “prevailing wind patterns” or similar expressing being involved or mentioned in those two references.
no comments
Author Response
Dear Reviewer,
Thank you so much for your constructrive comments to improvise our manuscript. We have made corrections accordingly:
1) The 102 geocoordinates are referring to the occurrences of individuals/populations of orchids from the 56 species recorded in this study (line 208-209).
2) Thank you so much for the great suggestion. We really hope to elaborate it further in our next manuscript. In this 3.2 heat maps, the occurrences of those narrowly distributed species are included, together with other more common or widespread species.
3) The statement was referred to Fay et al. (2015) after Chalk (2014) in “Orchid Conservation: Making the Links, Annals of Botany, Volume 116, Issue 3, September 2015, Pages 377–379, https://doi.org/10.1093/aob/mcv142, citating [29] in their article.
Reviewer 2 Report
The manuscript deserves to be published but presentation needs to be much improved. In a few places text should be re-phrased/re-worded to make it clearer. Some of the words used are highly unusual; sometimes you have also used wrong words. Below I have listed my comments.
--- There is some unnecessary repetition, the text should more concise.
line 3 Title: ….. Karst Hills of North Peninsular Malaysia; is this OK? I would have said 'northern'
16 'resulted in a discovery' should read 'resulted in the discovery'
29-30 'The obscure species that could be new to science': it should be said that this applies to all orchids, not just the ones that are new to science; and other plants too. I would say 'Orchids as well as other plants are threatened by anthropogenic activities such as quarrying and forest clearing for agriculture'.
37 'lies in the north Peninsular Malaysia' should read 'lies in the northern part of Peninsular Malaysia'
39 'surrounded' is the wrong word; should be 'includes' or something like that
40 'occupied' should read 'occupy'
42 'has stand out to be the focused area' should read 'stands out to be the focus area'.
42-43 'many geological studies especially on their limestone hills.' -- literature should be given.
44 limestone is misspelt
53-54 Paphiopedilum niveum and Cheirostylis goldschmidtiana: botanical author names need to be given here.
61-62 'whilst nearly 60% of the Sarawak’s flora are found on limestones'; actually not really relevant
66 'dense floral reports' --- what is this?
74 'body literatures'?
74 'works' should read 'work'
86 'decreased' should be 'decrease'
91-92 'The epiphytic orchids contribute about 40% of the total abundance and approximately 30% of orchid species richness in Malaysia' --- This is hard to believe. In other tropical countries the epiphytic orchids account for about two thirds (ca 66%) of all orchid species.
110 The Singgora Range should be labelled in Fig. 1
126 All of the 4 cited publications used for the identification are very old and partly outdated, especially the publication by Seidenfaden & Smitinand (see my comment below)
130-131 'and collection of living specimens is to be circumvented in any way possible.' But sometimes this is necessary, especially in sterile specimens or for ex-situ conservation. Besides this is somehow a contradiction to what is said a few lines lower down. I would leave this out.
153 Spathoglottis hardingiana was published by C.S.P.Parish & Rchb.f. (you have it correct in Table 1)
Table 1 It needs to be said clearly that 'Distribution' means 'Distribution in Peninsular Malaysia'
Table 1 Authors: please check the botanical author names, some are incorrect
Bulbophyllum meson was published by J.J.Verm., Schuit. & de Vogel
Corymborkis veratrifolia (Reinw) Blume: Reinw. is an abbreviation. There must be a dot at the end
Eria javanica (Sw.) Blume: very recently shown to be a misapplied name, and the name E. stellata is used for this species. This taxonomic change is probably too recent to be reflected in POWO
Geodorum terrestre (L.) Garay: this name was also misapplied by several authors. According to POWO (accessed on 8 May 2023) this name is a synonym of Calanthe amboinensis from central and eastern Malesia and the Pacific. Which plant do you mean?
169-170 Table 2 should be included in Table 1. Out of 56 species 53 are NE (not evaluated); to have a table with one column mostly with NE looks very strange. Alternatively, the three that have been evaluated could also be discussed in the text; there would be no need for a table.
199 Trias italicised
203 Trias intermedia was published in 1965, not 1924
202 His name is Tem Smitinand
203-207 This sounds as if this was a straight-forward revision. Please rephrase this, according to what Jaap Vermeulen, André Schuiteman and Eduard de Vogel said in the introduction of their 2014-paper. They also say that it is supported by DNA studies.
211 'high resemblances in plant size' ?? Surely this cannot be used as a diagnostic character
213 brachystachys is misspelt
219 after 108 years ??
221 'hyper endemic' should read 'narrow endemic'
218-227 But if Rogier van Vugt found it in 2010, then you cannot claim to have rediscovered it after more than one hundred years
236 Spathoglottis hardingiana: confined to Pulau Langkawi. No, the type is from Myanmar and it is also found in Bangladesh and Thailand.
242 Thai's should be Thailand's.
303 'accounted', should be 'observed here'
316 delete one of the two 'in'
330 References: General: author, please re-check the titles of publications and journals
349 'The Orchids of Thailand a Preliminary List' should read 'The Orchids of Thailand ‒‒ a Preliminary List'. This is a 4-volume publication, the date is 1959‒1965 (the date which you are giving, 1959, is the publication date of the first volume). But this publication should not be used for identification as it is completely out of date. Seidenfaden himself has revised most Thai orchid genera between 1965 and 2000.
353 POWO: Plants of the World Online; the first letter of World and Online capitalised
English language needs to be improved. In a few places text should be re-worded; some of the words used are highly unusual.
Author Response
Dear Reviewer,
Thank you so much for your constructive comments for the betterment of our manuscript. We really appreciate it. We have made changes accordingly based on your suggestions (on the manuscripts). Attached are some of the responses to your suggestions:
1) Line 42-43: References added.
2) Line 61-62: Irrelevant statement has been removed.
3) Line 74: 'Body literatures' have been changed as 'main references'.
4) Line 91-92: The statement has been removed.
5) Line 110: The location of Singgora Range has been added into Figure 1.
6) Line 126: These four cited references are the prominent text books/reference books for the identification of orchids in P. Malaysia and Thailand. We used these books for first stage of identification, and later cross-checked with much recent checklists prepared (added in manuscript) by various authors and readily available database (such as POWO).
7) Line 130-131: The sattement is rephrased as "Living collections of sterile specimens for further identification, ex–situ conservation purposes and germplasm studies were cultivated in the orchidarium, School of Biological Sciences, Universiti Sains Malaysia."
8) Table 1: We retained the accepted name of Eria javanica (Sw.) Blume following POWO.
9) Line 169-170: Table 2 has been included into Table 1.
10) Line 211: "high resemblances in plant size" has been detailed out as "high resemblances in vegetative characters".
11) Line 218-227: The statement is removed from manuscript. Rogier van Vugt published photographs of the orchid, but without any scientific publication.
12) Line 236: Added “in Malaysia” to clarify the species being known only on the island of Kedah (Pulau Langkawi).
Thank you so much!
Reviewer 3 Report
The article is relevant and reasonably well drafted. It contains important information. Having read the text, I would like to make some comments on how the manuscript could be improved.
1. I would suggest that Figure 1 be supplemented by an indication of adjacent regions or other geographic entities that border the study area.
2. Table 2 shows the species conservation assessment according to the IUCN criteria, but the methodology does not describe how the assessment was performed. The results of the assessment should also be indicated, according to which criteria the species is assigned to a particular category. On the other hand, is it appropriate to have a separate table if the majority of species are not evaluated (NE)? Maybe it would be sufficient to write in one sentence which species are assigned to a certain category and which are not evaluated.
3. I suggest changing the title of subsection 3.2. The results section 3.2 is now titled according to the method of presenting the research, but does not indicate the main idea of the sub-section or the question to be addressed.
4. The legends of the maps in Figure 2 are too small and almost impossible to read.
5. I would suggest adjusting the presentation style throughout the paper. In my opinion, it would be better not to emphasise in the text that there is a certain figure that contains certain information, but to provide information and a link to the figure. The current version of the text gives the impression that the illustration is the most important element, but not the results obtained or the discussion presented.
I suggest editing the style of the text. The language needs little editing.
Author Response
Dear Reviewer,
We thanked you for the constructive comments for the betterment of our manuscript. We really appreciate it. We have made corrections accordingly:
1) Figure 1 has been improvised.
2) Table 2 has been included as one in Table 1.
3) The title for subsection 3.2 has been changed to as "Heat map distributional data on species occurrences" to better reflect the main idea of this work.
4) The legends have been resized the best we could, because they are automatically run by the software.
5) We have reworked on the presentation style of the manuscript as suggested.
Thank you so much!
Reviewer 4 Report
This is an interesting manuscript showing a significant improvement in the knowledge of the orchid flora in an Asiatic diversity hot spot. I think the manuscript could have some improvements prior to its publication. I believe that Tables 1 and 2 could be fused into a single 1; especially considering that only two orchid species were assessed through the IUCN criteria. I would also like to call your attention to this recent article. This article presents some important insights that can enrich the discussion. For instance, orchid hot spots are highlighted as well as geographical areas deserving more attention and effective protection. Other relevant issues are discussed. For example, only a fraction (less than 24%) of the known Orchidaceae are globally preserved ex-situ in living collections or germplasm banks. Are you aware if part of your orchids are represented in such collections? This kind of information could render the text more attractive for a wider audience.
Vitt, P., Taylor, A., Rakosy, D., Kreft, H., Meyer, A., Weigelt, P., & Knight, T. M. (2023). Global conservation prioritization for the Orchidaceae. Scientific Reports, 13(1), 6718.
English (especially in the Abstract and Introduction) needs moderate editing.
Author Response
Dear Reviewer,
Thank you so much for positive suggestions to improvise our manuscript. We really appreciate it. We have made corrections accordingly:
1) Table 2 has been included in Table 1 as one.
2) We have included discussion on limestone habitat and population of orchids within it deserving more attention and effective protection as suggested.
Thank you so much!
Round 2
Reviewer 2 Report
The manuscript is vastly improved now. But I have 4 comments.
Table 1, line 15: If you really think you have found Calanthe amboinensis, then it must be clearly said that this is a new distribution record. [according to POWO, the species is found in central and eastern Malesia] [in POWO it is listed under the genus Phaius, but this genus has since been included in Calanthe]. Is your ID correct?
Paragraph 206-226: there is still no mention of molecular evidence. Please check the Vermeulen et al. paper and add it (if they really say this).
227-236: But if Rogier van Vugt found it in 2010, then you cannot claim to have rediscovered it in your study
360: 'The Orchids of Thailand ‒‒ a Preliminary List': This is a 4-volume publication, the date is 1959‒1965 (the date which you are giving, 1959, is the publication date of the first volume). But this publication is completely out of date. Seidenfaden himself has revised most Thai orchid genera between 1965 and 2000, and these can be used.
linguistic improvement required
Author Response
Dear Reviewer,
Thank you so much for your valuable insights in this second revisions. We really appreciate your thorough perusal and contributions in this second round. We have made corrections accordingly based on your suggestion:
1) Table 1, line 15 (Calanthe amboinensis). The species which we examined and recorded was Geodorum terrestre (L.) Garay, and native to P. Malaysia. It was listed in Seidenfaden and Wood (1992), and the checklist by Ong et al. (2017). When we verified the accepted name for this species in POWO, it was then treated as a synonym to Phaius amboinensis Blume. We admitted that we have mistakenly wrote it as Calanthe amboinensis in our manuscript. Thank you so much for pointing this out, or else we have reported a wrong distribution record. We have relisted this species in Table 1, Line 43 as Phaius amboinensis Blume. In POWO - Phaius is accepted as a separate genus from Calanthe, tentatively accepted awaiting further work on Calanthe (as proposed by Chase et al. (2020) in Taxon (69:6 - 2782).
https://www.mybis.gov.my/sp/63019
https://powo.science.kew.org/taxon/urn:lsid:ipni.org:names:999061-1
2) Molecular evidence and related references were included in the paragraph (253-259):
However, taxonomic revision supported by morphological examinations and DNA evidences have generated new combinations in Bulbophyllum, formally transferring Trias together with several other satellite genera to Bulbophyllum. This transfer was made due to the similarities of vegetative structures and flower attributes shared by species among the two genera. Phylogenetic analysis using two plastid genes, rbcL and matK; and one nuclear region nrITS has shown that all species of Bulbophyllum in section Trias were embedded among other groups of Bulbophyllum [25][24] Vermeulen, J.J.; Schuiteman, A.; de Vogel, E.F. Nomenclatural changes in Bulbophyllum (Orchidaceae; Epidendroideae). Phytotaxa 2014, 166(2), 101-113.
[25] Wonnapinij, P.; Sriboonlert, A. Molecular phylogenetics of species of Bulbophyllum Trias (Orchidaceae; Epidendroideae; Malaxidae) based on nrITS and plastid rbcL and matK. Phytotaxa 2015, 226(1), 1-17.
3) Line 271: We have changed the word 'rediscovery' to 'recollection'.
4) The references has been updated to:
[11] Seidenfaden, G.; Smitinand, T. The Orchids of Thailand – a Preliminary List, Part 4; The Siam Society: Bangkok, Thailand, 1965.
Thank you so much!
Reviewer 3 Report
The revised article has been significantly improved in line with the reviewers' comments, and I now have no substantive comments on the text. The authors have taken into account the comments made in my previous review.
Minor editorial revisions required.
Author Response
Dear Reviewer,
Thank you so much for your kind suggestions for the improvement of our manuscript. We really appreciate it!
Cheers!